# CD14 Is Involved in the Interferon Response of Human Macrophages to Rubella Virus Infection

**DOI:** 10.3390/biomedicines10020266

**Published:** 2022-01-26

**Authors:** Erik Schilling, Lukas Pfeiffer, Sunna Hauschildt, Ulrike Koehl, Claudia Claus

**Affiliations:** 1Institute of Clinical Immunology, Medical Faculty, Leipzig University, Johannisallee 30, 04103 Leipzig, Germany; erik.schilling@medizin.uni-leipzig.de (E.S.); Ulrike.Koehl@medizin.uni-leipzig.de (U.K.); 2Institute of Medical Microbiology and Virology, Medical Faculty, Leipzig University, Johannisallee 30, 04103 Leipzig, Germany; lukaspfeiffer05@gmail.com; 3Institute of Biology, Faculty of Life Sciences, Leipzig University, Talstrasse 33, 04103 Leipzig, Germany; shaus@rz.uni-leipzig.de; 4Fraunhofer Institute for Cell Therapy and Immunology, 04103 Leipzig, Germany; 5Institute for Cellular Therapeutics, Hannover Medical School, 30625 Hannover, Germany

**Keywords:** chemokine, cytokine, extracellular flux analysis, glycolysis

## Abstract

Macrophages (MΦ) as specialized immune cells are involved in rubella virus (RuV) pathogenesis and enable the study of its interaction with the innate immune system. A similar replication kinetics of RuV in the two human MΦ types, the pro-inflammatory M1-like (or GM-MΦ) and anti-inflammatory M2-like (M-MΦ), was especially in M-MΦ accompanied by a reduction in the expression of the innate immune receptor CD14. Similar to RuV infection, exogenous interferon (IFN) β induced a loss of glycolytic reserve in M-MΦ, but in contrast to RuV no noticeable influence on CD14 expression was detected. We next tested the contribution of CD14 to the generation of cytokines/chemokines during RuV infection of M-MΦ through the application of anti-CD14 blocking antibodies. Blockage of CD14 prior to RuV infection enhanced generation of virus progeny. In agreement with this observation, the expression of IFNs was significantly reduced in comparison to the isotype control. Additionally, the expression of TNF-α was slightly reduced, whereas the chemokine CXCL10 was not altered. In conclusion, the observed downmodulation of CD14 during RuV infection of M-MΦ appears to contribute to virus-host-adaptation through a reduction of the IFN response.

## 1. Introduction

The innate immune response is an important first line of defence against virus infections and a regulatory component for the adaptive immune response. As one of its cellular components, macrophages (MΦ) fulfil a broad range of functions and are present in multiple tissues and organs. The MΦ population in the body comprises mature tissue MΦ besides MΦ that are differentiated from monocytes, which are generated by haematopoietic stem cells in the bone marrow and released into the blood circulation [1]. Depending on their tissue environment and the respective tissue-derived signals MΦ can undergo classical (M1) or alternate (M2) polarization including intermediate subtypes [2]. The polarization state comprises differences in their functionalities and reliance on metabolic pathways: M1 MΦ are destructive on their surrounding tissue and rely mainly on glycolysis, whereas anti-inflammatory M2 MΦ function in tissue re-modelling and wound-healing and utilize mainly mitochondrial respiration [1,3]. However, based on these tissue-protective functions M2 MΦ also enable persistence of some pathogens in the respective tissue [4]. Such a close association of M2 MΦ with virus persistence applies to rubella virus (RuV) as a member of the family *Matonaviridae*. Under some immune privileged conditions reactivation of the vaccine strain of this single-stranded RNA virus with a genome in positive polarity can occur [5]. There is an association between RuV and MΦ as these cells play an important role in the replication and spread of RuV and as such in rubella pathogenesis. In line with this, van der Logt et al. showed in an early work that monocyte-derived MΦ support RuV replication [6]. Furthermore, the results of Lazar et al. revealed that in tissue samples from patients with fatal congenital rubella syndrome (CRS) mainly alveolar MΦ were positive for RuV antigen [7]. In our previous work we demonstrated that both human GM-MΦ and human M-MΦ were infected with RuV and replication of RuV occured at a similar rate in both cell types [8]. RuV infects only humans and until 2020 no close relatives of RuV among animal and human viruses were known. Through metagenomic analysis ruhugu and rustrela virus were identified in different animal species including bats and rodents [9]. The high sequence similarity between RuV and ruhugu and rustrela virus highlights their potential as emerging pathogens with the possibility of transmission to humans [9]. In addition to the discovery of its close relatives, an association of RuV with granulomas in patients with primary immunodeficiencies (PIDs) [10] re-emphasizes the importance of research into the interaction of RuV with immune cells. The RuV vaccine RA27/3 was licensed in 1969 and is safe and very efficient with long lasting immunity [11]. However, in recent years a close correlation between reactivated RuV vaccine with granulomas in PID patients was shown [10]. Within these granulomas, RuV antigen was mainly identified in keratinocytes and M2 MΦ. The persistence of RuV in M2 MΦ in granulomas of PID patients suggests its low cytopathogenic impact along with the possibility of adaptation to antiviral mechanisms.

The functional state of MΦ can be directed by multiple factors. Among those the monocyte differentiation antigen CD14 serves as an innate immune receptor and fulfils several functions. CD14 promotes inflammation and its expression on tumour-associated immune cells influences tumour immunosurveillance [12]. Moreover, CD14 functions as a pattern recognition receptor (PRR) and as a co-receptor for several toll-like receptors (TLRs) [13]. Ligands of CD14 comprise not only bacterial and viral pathogen-associated molecular patterns (PAMPs), but also endogenous molecules. The release of heat shock protein 70 (HSP70) activates pro-inflammatory signals in a CD14-dependent manner [14]. As an example for a bacterial PAMP, CD14 binds lipopolysaccharide (LPS) from gram-negative bacteria and transfers it to TLR4 [15,16]. Among viral PAMPs recognized by CD14 are the fusion protein of respiratory syncytial virus and components of cytomegalovirus (CMV) particles in association with TLR4 and TLR2, respectively [17,18].

Here we addressed the involvement of CD14 in RuV-induced innate immune response in M1- and M2-like human MΦ, which can both be productively infected by RuV [19]. The here utilized M1-like (GM-MΦ) and M2-like (M-MΦ) phenotype was based on in vitro differentiation of PBMC-derived monocytes in the presence of GM-CSF and M-CSF, respectively. After RuV infection, a reduction of CD14 occurred in both MΦ types at mRNA and protein level, which was especially prominent in M-MΦ. The application of exogenous interferon (IFN)-β indicated that this reduction was not related to the RuV infection-induced IFN response. However, the application of extracellular flux analysis together with a glycolysis stress test revealed that the loss of glycolytic capacity noted after RuV infection was also induced by exogenous IFN-β. As a mimetic of the reduced CD14 expression level during RuV infection, we added anti-CD14 blocking antibodies prior to RuV infection. In comparison to the isotype control, blockage of CD14 increased the amount of extracellular virus particles and altered specific components of the cytokine/chemokine profile after infection with RuV: type I and III IFNs were significantly and TNF-α was slightly reduced, whereas CXCL10 (IP-10) was not affected. With our study we highlight that CD14 is involved in the immune response of MΦ against RuV. The RuV infection-associated reduction in CD14 expression could support its course of infection in MΦ and potentially contribute to the in vivo persistence of RuV in M2 MΦ. 

## 2. Materials and Methods

### 2.1. Reagents and Antibodies

CD14 blocking antibody (Ab) (MEM-18) and the corresponding mouse IgG1 isotype control Ab (MOPC-21) were obtained from EXBIO Praha (Vestec, Czech Republic). Recombinant human IFN-β was purchased from Peprotech (Hamburg, Germany). 

### 2.2. Cell Separation and Cell Culture

Human peripheral blood mononuclear cells from buffy coats of healthy donors (blood service, Institute of Transfusion Medicine University Hospital Leipzig; ethics license 272-12-13082012) were isolated using a Ficoll-Paque Plus (GE Healthcare, Little Chalfont, Buckinghamshire, UK) centrifugation step. Following a washing step with PBS containing 0.3 mM EDTA, monocytes were isolated by counter-flow elutriation using the JE-5.0 elutriation system (Beckman Coulter, Brea, CA, USA), as described previously [20].

The purity of isolated monocytes was determined by staining of CD14 surface marker with anti-CD14-APC mouse Ab (M5E2, BioLegend, San Diego, CA, USA) followed by flow cytometry analysis. Monocyte fractions with a purity of at least 90% were suspended (5 × 10^5^ cells/mL) in RPMI 1640 medium supplemented with 10% (*v*/*v*) fetal calf serum (FCS, Sigma-Aldrich, Taufkirchen, Germany), 100 U/mL penicillin and 100 mg/mL streptomycin (both from Seromed Biochrom KG, Berlin, Germany). Differentiation into MΦ was induced by adding 500 IU/mL GM-CSF (Leukine, sargramostim) (GM-MΦ) or 50 ng/mL M-CSF (R&D Systems, Minneapolis, MO, USA) (M-MΦ). After 7 days of incubation at 37 °C and 5% CO_2_ in teflon bags (Zell-Kontakt, Nörte-Hardenberg, Germany; fluorinated ethylene propylene foil, 50 µm, hydrophobic), MΦ were harvested and counted. Prior to use in experiments, MΦ (5 × 10^5^ cells/mL) were suspended in RPMI 1640 medium (with L-glutamine, 25 mM HEPES and phenol red, GE Healthcare (Little Chalfont, Buckinghamshire, UK) supplemented with 10% (*v*/*v*) FCS, 100 U/mL penicillin and 100 mg/mL streptomycin and plated followed by incubation for 2 h in cell culture plates.

### 2.3. Flow Cytometry Analysis 

Flow cytometry analysis of markers expressed at the surface of MΦ was performed with direct dye labelled antibodies anti-CD14-APC mouse Ab (M5E2, BioLegend), anti-CD40-PerCp mouse Ab (Elabscience Biotechnology, Houston, TX, USA), anti-CD80-PE mouse Ab (L307, BD Pharmingen, San Jose, CA, USA), anti-CD86-pacific blue mouse Ab (IT2.2, BioLegend) and the respective isotype controls as described previously [21].

### 2.4. Virus Infections of MΦ

Infection of MΦ with RuV (in agreement with ethics license 001/19-ek) was carried out with the low-passaged (up to passage 10) clinical isolate RVi/Wuerzburg.DEU/47.11_12-00009 (Wb-12). RuV was generated on epithelial Vero cells (green monkey kidney, ATCC CCL-81). Dulbecco’s modified Eagle’s medium (DMEM; Thermo Fisher Scientific, Waltham, MA, USA) supplemented with 10% FCS was used for cultivation of Vero cells. As titered on Vero cells by standard plaque assay for RuV, human MΦ were infected at an MOI of 1.5 if not indicated otherwise. As a control, RuV was inactivated by UV-light with 900,000 μJ/cm^2^ (UV-Stratalinker 2000, Stratagene, San Diego, CA, USA). Before infection with virus stock solution, the culture media was discarded. After 2 h virus stock was removed and fresh culture media was added after one washing step with PBS.

### 2.5. Immunofluorescence Microscopy and Apoptosis Assessment

The immunofluorescence analysis of RuV-infected MΦ using monoclonal antibody against RuV capsid protein (clone 2–36, Meridian Life Science, Inc., Memphis, TN, USA) was performed as described previously [8]. For apoptotic assessment the CellEvent Caspase-3/7 Green Detection Reagent (Invitrogen, Thermo Fisher Scientific). Cleavage of the DEVD peptide by activated caspase 3/7 results in release of the conjugated nucleic acid-binding dye. Upon cleavage the dye becomes fluorescent and can be detected in the nucleus. The proportion of fluorescent nuclei was determined by fluorescent microscopy.

### 2.6. Measurement of Cytokines/Chemokines in Culture Supernatants

TNF-α and IFNs were determined in culture supernatants of MΦ (5 × 10^5^/mL) using a customized version of LEGENDPLEX human type 1/2/3 Interferon panel (5-plex) kit in combination with human TNF-α beads B3 (BioLegend) according to the manufacturer’s protocol. CXCL10 protein in culture supernatants of MΦ (5 × 10^5^/mL) was measured using a human CXCL10 ELISA kit (R&D Systems, Minneapolis, MN, USA) according to the manufacturer’s protocol.

### 2.7. Western Blot 

Analysis of protein expression by Western blot was performed as described previously [8]. Briefly, the following antibodies were used at indicated dilutions for the detection of proteins on PVDF membranes: anti-CD14 mouse Ab (D7A2T, 1:1000); anti-RuV (E1) mouse Ab (1:500, Merck, Darmstadt, Germany); anti-phospho-Stat-1 rabbit Ab (Tyr701, D4A7, 1:1000) and anti-β-actin mouse Ab (AC 74, 1:2000, Sigma-Aldrich, St. Louis, MO, USA). The POD-conjugated secondary antibodies goat anti-rabbit IgG Ab (1:20,000, Dianova, Hamburg, Germany) or goat anti-mouse IgG Ab (1:8000, Sigma-Aldrich, St. Louis, MO, USA) were used for the detection of the respective primary antibodies. 

### 2.8. RNA Isolation and Reverse Transcription

The RNeasy Mini Kit (Qiagen, Hilden, Germany) was used according to the manufacturer’s protocol for total RNA extraction from MΦ (3 × 10^5^) after washing with PBS and including DNase I treatment. Reverse transcription of equal amounts of RNA (250 ng) to cDNA was performed as previously described [20].

### 2.9. Real-Time PCR (qPCR)

Real-time PCR was carried out as described previously [8] with the following primers: GNB2L1 (Forward 5′-GAGTGTGGCCTTCTCCTCTG-3′; Reverse 5′-GCTTGCAGTTAGCCA GGTTC-3′) [22], IFN-β (Forward 5′-AACTTTGACATCCCTGAGGAGATTAAGCAG-3′; Reverse 5′-GACTATGGTCCAGGCACAGTGACTGTACTC-3′) [23], IFN-λ1 (Forward 5′-GCAGGTTCAAATCTCTGTCACC-3′; Reverse 5′-AAGACAGGAGAGCTGCAACTC-3′) [24], TNF-α (Forward 5′-TCAGCCTCTTCTCCTTCCTG-3′; Reverse 5′-GGCTACAGGCTTGTCACTCG-3′) [25], CD14 (Forward 5′-GCCGCTGTGTAGGAAAGAAG-3′; Reverse 5′-GCTGAGGTTCGGAGAAGTTG-3′) [26], CXCL10 (Forward 5′-CCATTCTGATTTGCTGCCTTA-3′; Reverse 5′-TGATGCAGGTACAGCGTACAG-3′) [27].

### 2.10. Metabolic Assessment of the Extracellular Acidification Rate (ECAR) through Extracellular Flux Analysis 

Extracellular flux measurement was carried out with an XFp analyser (Agilent Seahorse Technologies, Santa Clara, CA, USA). XFp Seahorse plates were seeded with MΦ at a density of 4 × 10^4^ cells per well. Before measurement culture medium was replaced by XF base medium (Agilent Seahorse Technologies) supplemented with 2 mM glutamine. The glycolysis stress test kit (Agilent Seahorse Technologies) was carried out as described by the manufacturer. As exemplified in Appendix A, 12 measurement points covered successive injection of glucose, oligomycin (1 µM) and 2-deoxy-glucose. 

### 2.11. Statistical Analysis

Data are shown as the mean ± standard deviation (SD). All statistical analyses were carried out using the SigmaPlot^®^ software. Statistical significance was calculated with Student’s *t*-test or ANOVA test. 

## 3. Results

### 3.1. The Initial Infectious Dose Has a Slight Influence on RuV Infection in GM- and M-MΦ 

In our previous study, we examined the response of MΦ to RuV through the application of just one infectious dose. However, susceptibility of a given cell line can be influenced by variations in the applied infectious dose. Especially under conditions of high cellular susceptibility, a low viral load is sufficient to initiate an efficient infectious cycle. Thus, we included a slightly higher and lower MOI in addition to the previously published MOI of 1.5 [8], namely an MOI of 0.5, 1.5, and 4.5. Figure 1A shows that the number of infected cells as determined by immunofluorescence analysis with anti-capsid antibodies was increasing on both MΦ types with a higher MOI. Hereafter, the time-dependent release of extracellular virus particles was studied under these three different MOIs. Except for 12 h post-infection (hpi) on GM-MΦ, similar amounts of virus particles were present after infection with different MOIs (Figure 1B). This could reflect a faster uptake rate or an earlier initiation of the replication cycle on M-MΦ. An increase in extracellular infectious particles by about one log-step occurred for both MΦ types and all applied MOIs between 12 and 24 hpi (Figure 1B,C). As a next step, we addressed RuV-associated apoptosis activation in MΦ. Although RuV displays in only on a limited number of cell lines a cytopathic effect [28,29], RuV infection activates an apoptotic response [29,30]. As an early marker for apoptotic events, we performed an assay with a cleavable and non-fluorescent DEVD peptide bound to a nucleic acid intercalating dye. This dye visualizes caspase-3/7 cleavage activity through fluorescent dye-positive signals in the nucleus, which were counted by fluorescence microscopy. At 24 hpi even after a high MOI the number of apoptotic cells did not exceed 10% of the total cell number (Figure 1D). At 48 hpi and after a high MOI of 4.5 the number of apoptotic cells in M-MΦ were with 18.7 ± 4.3% slightly higher than in GM-MΦ with 12.4 ± 4.8% (Figure 1D). Based on infection rate and apoptosis induction, we applied an MOI of 1.5 in subsequent experiments. In conclusion, a variation in the initial viral load revealed only slight variations in the course of infection, especially in the kinetics of the production of extracellular virus particles and in the activation of apoptosis. 

### 3.2. During RuV Infection of MΦs the Expression of the Pattern Recognition Receptor CD14 Is Reduced

CD14 is not only an innate immune receptor, but also a MΦ differentiation marker. Thus, we first compared its expression level on both MΦ types by flow cytometric analysis over time of their cultivation. At all analysed time points, the CD14 surface expression level was higher on GM- than on M-MΦ (Figure 2A). In agreement with previous reports [31,32], the CD14 surface expression level on both MΦ types dropped over the first 24 h of cultivation and remained at a comparable level between 24 and 48 h after plating (Figure 2A). Hereafter we addressed the CD14 expression level on both MΦ types after RuV infection. The similar course of infection of RuV on GM- and M-MΦ allows for such a comparative analysis, as identified differences are unlikely to be caused by differences in the infection kinetics. As an additional control, we also addressed the impact of UV-inactivated RuV (RuV^UV^) on CD14 expression, as MΦ are prone to sense pathogens. RuV^UV^ allows for identification of aspects that are associated with the recognition of the virion and in the absence of a productive replication cycle. The drop in CD14 surface expression over the first 24 h of MΦ cultivation (Figure 2A) was also reflected by a decrease in the mRNA expression in mock-, RuV^UV^- and RuV-infected GM-MΦ (Figure 2B) and M-MΦ (Figure 2C). In comparison to the mock control, RuV infection resulted in a reduction in CD14 mRNA expression in both, GM- and M-MΦ at 24 hpi (Figure 2B,C). Additionally, both MΦ types responded to RuV^UV^ through an increase in CD14 mRNA. On M-MΦ these changes induced by RuV and RuV^UV^ were significant in comparison to the mock-infected control. Accordingly, the RuV infection-associated impact on CD14 was also reflected by a significant reduction in CD14 in the total cell lysate from M-MΦ extracted at 24 hpi (Figure 2D). Endocytosed CD14 is degraded in lysosomes [33]. Thus, we hypothesized, that a low abundance of CD14 mRNA as seen for RuV-infected MΦ would not be able to replenish the CD14 pool at the surface. We addressed this aspect by flow cytometric analysis at 24 and 48 h to take the turn-over rate of CD14 into account. In comparison to the mock control, a decrease in CD14 surface expression was detected for both MΦ types with a significant reduction for M-MΦs at 48 hpi (Figure 2E). The impact of RuV on CD14 expression level is especially noteworthy, as MΦ infected with RuV^UV^ showed the opposite tendency of RuV: CD14 surface expression level was increased in comparison to the mock-infected cells (Figure 2D). Taken together, a productive RuV infection reduced CD14 expression. This impact was significant for M-MΦs, whereas GM-MΦ revealed a tendency similar to M-MΦ.

### 3.3. Glycolytic Reserve of MΦ Was Lost in the Presence of IFN-β

In response to RuV infection, GM- and M-MΦ generate type I and III IFNs [8,19]. The infection with RuV^UV^ induced no IFN-β production [8]. Our own published data shows that at 24 hpi IFN-β as an antiviral response to RuV infection was generated on GM- and M-MΦ at a concentration of 219 ± 170 pg/mL and 666 ± 199 pg/mL, respectively [8]. This is also illustrated in Appendix A, which indicates the generation of IFN- at a concentration of 466 ± 119 pg/mL on M-MΦ at 24 hpi. After medium change performed at 24 hpi IFN-β dropped to a concentration of 59 ± 38 pg/mL after an additional incubation of 24 h. In order to investigate which effects of the RuV infection on M-MΦ are associated with IFN-β production, exogenous IFN-β was applied to M-MΦ and compared to RuV infection. Subsequent IFN-associated signalling might alter the expression of surface receptors on MΦ and affect their metabolic state. Thus, we next addressed the influence of IFN-β through its exogenous application. First, we analysed CD40, CD80 and CD86 besides CD14 on RuV-infected and exogenous IFN-β-treated MΦ at an incubation interval of 24 and 48 h. CD40, CD80, and CD86 are markers for GM-MΦ and important for activation of the adaptive immunity. Furthermore, they can be activated by TLR signalling: upon TLR3-stimulation these markers were upregulated on murine and human M2 MΦ in an IFN-α/β-dependent manner, which rendered their phenotype more M1-like [34]. The analysis of their surface expression revealed similar and differential effects between RuV infection and exogenous IFN-β: CD40 and CD86 were induced under both conditions, whereas the downregulation of CD14 and the upregulation of CD80 were specific for RuV infection (Figure 3A). Thus, in the presence of RuV the expression of the GM-MΦ markers CD40, CD80, and CD86 were increased indicative for a shift in polarization characteristics. Under these conditions, the influence of RuV^UV^ infection on the surface expression of the examined markers was not significant in comparison to the mock control (Figure 3A). Second, the metabolic impact of RuV infection and exogenous IFN-β-treatment was examined. We focused on M-MΦ, as alterations in CD14 were more pronounced than in GM-MΦ (Figure 3A). For metabolic analysis extracellular flux analysis with the glycolysis stress test kit was used, as the activation of MΦ is known to be associated with an increase in glycolysis [35,36]. Appendix A reflects the terms glycolysis, glycolytic capacity and glycolytic reserve based on the successive injection of glucose, the ATPase inhibitor oligomycin and the glycolysis inhibitor 2-deoxy-glucose. Figure 3B(I) underlines that although after RuV infection glycolysis and glycolytic capacity did not shift significantly compared to the mock control; the glycolytic reserve changed significantly and was no longer measurable. Comparable to RuV infection of M-MΦ, co-incubation of mock- and RuV^UV^-controls with exogenous IFN-β (10 ng/mL) did also reduce glycolytic capacity and abolish glycolytic reserve (Figure 3B(II)). In summary, our data suggests that while the impact of RuV infection on CD14 surface expression appears to be independent from IFN-β response, the RuV-induced alteration of glycolytic activity in M-MΦ was highly similar to metabolic inhibition by IFN-β.

### 3.4. Blockage of CD14 Prior to RuV Infection of M-MΦs Reduces the Associated IFN Response

CD14 is a surface receptor and serves as a marker during differentiation of monocytes into MΦs [37]. Antibody-mediated blockage of CD14 is a feasible approach to study its function. Blockage results directly in loss of CD14 receptor functions, such that expression of CD14 can be maintained during in vitro differentiation of monocytes [21]. Thus, we addressed the role of CD14 during RuV infection through the application of blocking antibodies to M-MΦs, as CD14 was significantly reduced on this MΦ subtype after RuV infection (Figure 2). As expected, Western blot analysis reflects a significant reduction in CD14 through blocking antibodies in comparison to the isotype control (Figure 4(AI,AII)). Moreover, under either condition, blocking antibodies or isotype control, the expression of CD14 in RuV-infected M-MΦs was lower than in the mock controls (Figure 4AI). According to the application scheme shown in Figure 4, anti-CD14 antibodies were applied prior to infection with RuV. Thereafter the generation of extracellular virus particles as an indicator for virus replication and particle production efficiency was determined. In agreement with a slightly increased amount of viral E1 protein after application of anti-CD14 blocking antibodies, extracellular virus particles were increased compared to the isotype control (Figure 4B). As a next step we addressed, whether blockage of CD14 alters glycolytic activity as it was noted after RuV infection and after application of exogenous IFN-β (Figure 3B). Figure 4C reveals a similar glycolytic activity between blocking antibodies and isotype control. Thus, blockage of CD14 did not alter metabolic activity. As metabolic activity appeared not to be influenced, we addressed the inflammatory response through analysis of the chemokine/cytokine response. Notably, the application of anti-CD14 blocking antibodies significantly reduced the mRNA expression level of IFN-β and IFN-λ1 and TNF-α, whereas the chemokine CXCL10 (IP-10) was not affected (Figure 4D). This reduction was also present at the protein level: compared to the isotype control, a multiplex bead-based assay revealed a significant reduction of IFN-β and IFN-λ1 in the supernatant of RuV-infected M-MΦ with application of anti-CD14 antibodies (Figure 4E) and tendency of reduction for TNF-α, whereas CXCL10 was not altered (Figure 4F). Collectively our data point toward a CD14-dependent expression of IFNs during RuV infection of MΦ and a rather pro-viral involvement of the RuV infection-associated decrease in CD14 expression. 

The main conclusions of our manuscript are summarized in Figure 5.

## 4. Discussion

Virus infection-associated changes of MΦ functions contribute to downmodulation of immune functions during virus infections [39]. Thus, the investigation of these changes could support our understanding of viral pathogenesis. The persistence of reactivated RuV vaccine-derived viruses in M2 MΦ in patients with PID [40] suggests a contributory role of this innate immune cell type to rubella pathogenesis, which might even represent a possible cellular reservoir for RuV. The clinical finding on maintenance of vaccine-derived RuV in M2 MΦ is in line with our observations in this study on the reduced expression of CD14 during RuV infection, which was more pronounced in M (M2-like)- than in GM (M1-like)-MΦs. This appears to be part of virus infection-induced alterations to support virus replication as shown here through a reduced IFN response and an increase in number of extracellular virus particles after antibody-mediated blockage of CD14. 

CD14 itself is a PRR and serves as a co-receptor for several TLRs, including TLR4 at the plasma membrane as well as TLR3 at the plasma membrane and in the endosome and as such CD14 fulfils multiple functions [41]. While TLR3 serves as a PRR for nucleic acids [42], TLR4 is involved in the recognition of bacterial components such as LPS, but also of viral glycoproteins as shown for Ebola virus glycoprotein [43] and the F-protein of respiratory syncytial virus [17]. The RuV PAMPs and the receptors involved in their recognition are still not known. Nevertheless, human nucleotide polymorphisms (SNPs) that are associated with variations in the immune response to rubella vaccination provide some indications. The cytokine response to RuV vaccine strain including the generation of IFN-γ, TNF-α, and GM-CSF were affected by SNPs in the IFN-α/β receptor (IFNAR2) and the TLR3 and TLR4 genes [44,45,46]. Thus, the identified SNPs could be involved in a differential immune response to rubella vaccination in humans. Namely, two SNPs within the TLR3 gene were accompanied by a reduced secretion of GM-CSF [46]. In addition to the contributory role of the PRRs TLR3 and TLR4, these studies also highlight the importance of innate immune responses for the overall antiviral countermeasures against RuV. The generation of dsRNA during RuV infection appears to be the main activating signal for an IFN response as RuV^UV^ does hardly induce IFNs [8]. The relevance of the expression of the PRR CD14 on MΦs for inflammatory processes is also reflected by physiological processes: absence of CD14 on resident intestinal MΦs adds to the low level of inflammation within the intestinal mucosa in the presence of gut commensal microbiota [47]. The differential impact of RuV and RuV^UV^ on CD14 expression and the rather low level of IFN induction after application of RuV^UV^ [8] might be due to the activation signal imposed by viral RNA and dsRNA as replicative intermediates of the RuV productive replication cycle, which is not present during RuV^UV^ infection. In line with this hypothesis, the induction of type I IFNs and pro-inflammatory cytokines by cell-free Ebola virus-like particles (eVLP) was synergistically enhanced after its co-application with Poly I:C as a dsRNA analogue, which is active through the TLR3 or RIG-I pathway [48]. Moreover, dsRNA as an activator of TLR3 can be released from virus-infected cells and as such contribute to the induction of an inflammatory response [49].

The role played by CD14 in host defence mechanisms is always protective through activation of an immune response [41], whereas the level of CD14 expression in MΦs during infection depends on the type of the virus and bacterium. Comparable to infection with RuV, the expression of ORF3 of hepatitis E virus (HEV) reduced CD14 expression [50]. In the case of human monocyte-derived MΦ CMV infection is associated with a maintained high expression level of CD14 as compared to the decline in mock infection over time of cultivation [51]. The maintenance of CD14 expression during HCMV infection was discussed to fuel their pro-inflammatory response and thus contribute to the pathogenesis of HCMV through induction of inflammation in infected tissues [51,52]. By implication, this supports our hypothesis on the contribution of a reduced CD14 expression level to maintenance of RuV in M-MΦs on a rather low inflammatory background as shown here through a reduced IFN expression level after blockage of CD14 prior to RuV infection. The link between CD14 expression and IFN response is also supported by literature data on the infection of murine MΦs with VSV. The VSV glycoprotein activates the TLR4/CD14 complex and as such the antiviral type I IFN pathway [53]. In addition to the observed reduction in CD14 after RuV infection, a slight shift to a more GM-MΦ phenotype as indicated through the increase in the GM-MΦ markers CD40, CD80, and CD86 could also be associated with a lower IFN response. The RuV-associated IFN response on GM-MΦ is lower than on M-MΦ [8].

Based on our data we hypothesize that CD14 is involved in the IFN response during RuV infection. An open question for future research is how viruses such as RuV could modulate CD14 expression and how CD14 contributes to the antiviral response of MΦs against RuV. CD14 lacks an intracellular domain and as such CD14 on its own is not able to transmit intracellular signals upon binding of its respective ligand. CD14 was proposed as a transporter of various ligands through the cells and was grouped as a member of the TAXI proteins, the transporters associated with the execution of inflammation [33]. Potential cofactors of CD14 as outlined hereafter include TLR3 and TLR4. As a follow-up of the study by Okumura and colleagues on the activation of TLR4 through eVLP [54], the associated signalling cascade was analysed by Ayithan and colleagues through activation of dendritic cells (DC) with eVLP [48]. Whereas the stimulation of DCs from MyD88^−/−^ mice had no impact on IFN-β expression and a slight effect on TNF-α was noted, whereas in DCs from TRIF^−/−^ mice IFN-β was almost absent and TNF-α was reduced. TLR3 recognizes dsRNA and signals through TRIF, whereas the other members of the TLR family signal through MyD88 [55,56]. Further line of evidence supports the notion on the involvement of CD14 in TLR3-signalling, including the proposed interaction of CD14 with the dsRNA analogue Poly I:C in support of the activation of TLR3 [57]. This was further addressed by Rajaiah and colleagues through the application of Poly I:C to bone marrow-derived macrophages from CD14^−/−^ mice. The generation of TNF-α was greatly reduced, whereas IFN-β was slightly reduced and CXCL10 was not altered, which suggests that some effects of TLR3 are CD14-independent or partially CD14-dependent [58]. With our data we support the notion on the involvement of CD14 in the generation of IFNs by human MΦs in response to a virus infection. 

Moreover, there are multiple points of interaction between cellular metabolism and antiviral innate immune response [59]. Our data suggests that the impact of RuV infection of the metabolic activity of M-MΦs is mainly related to the presence of IFN-β. With our data on human MΦs during a virus infection we complement recent data on the involvement of IFN-β in the metabolic response of murine bone marrow-derived macrophages to infection with Mycobacterium tuberculosis and the so far undercharacterized role of type I IFNs to the metabolic response of MΦs [35]. Comparable to our observations, IFN-β induced a loss of glycolytic reserve in murine MΦs [35]. Accordingly, the modulation of IFN-β expression through alteration in CD14 expression levels could support RuV infection through a reduced impact of IFN-β on glycolysis and as such an enhanced availability of metabolites and energy. Notably, in contrast to Mycobacterium tuberculosis infection of murine MΦs [35], basal glycolysis during RuV infection of human MΦs was slightly higher than after application of exogenous IFN-β to control infections. Moreover, the antiviral activity of IFNs includes the induction of IFN-induced genes (ISGs), which encode antiviral proteins that interfere with various steps of the viral growth cycle. This antiviral effect by IFN-β against RuV was shown in a previous study by our own group through exogenous application of IFN-β to A549 prior RuV infection [19]. 

In conclusion, our study provides further evidence in support of a contributory role of CD14 to sensing of viral infections besides its known high relevance for the recognition of bacterial components. The downmodulation of CD14 appears to be supportive for RuV infection of MΦs and could potentially be involved in RuV maintenance, as it was shown in association with M2 MΦs in granulomas of PID patients [40]. The identification of high serum soluble CD14 level in COVID19 patients highlights its potential as a therapeutic target [60,61]. A CRISPR/Cas9 screen identified genes involved in downmodulation or upregulation of CD14 and as such pharmacological compounds that target these processes [61]. Furthermore, reactivation and persistence of RNA viruses such as RuV holds many open questions including the potential cellular reservoir. Through their access to various tissue and their involvement in tissue homeostasis MΦs could be modified during virus infections to support viral persistence. Our data adds novel aspects through the involvement of CD14 in the response of human MΦs to RuV. 

## Figures and Tables

**Figure 1 biomedicines-10-00266-f001:**
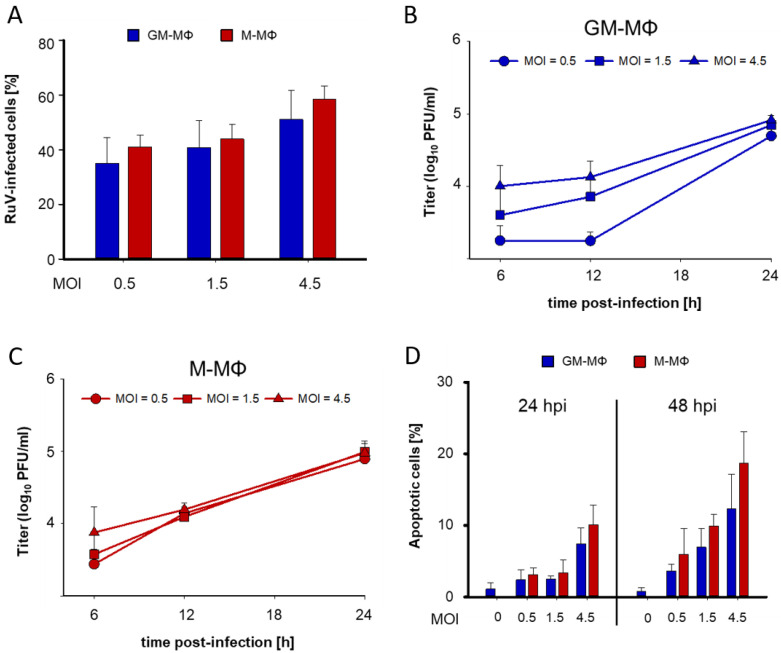
A higher initial infectious dose in GM- and M-MΦs influences early release of extracellular virus particles and apoptosis induction. After infection of GM- and M-MΦ with three different MOIs the course of infection was analysed. (**A**) The number of RuV-positive cells was determined at 24 hpi through immunofluorescence analysis with anti-capsid antibodies. (**B**,**C**) The number of extracellular virus particles was determined by standard plaque assay. (**D**) Apoptotic cells were counted on a fluorescent microscope as cells with CellEvent Caspase 3/7 green reagent-positive nuclei in reference to the total cell number. (**A**,**D**) Two random fields per microscope slide were counted (*n* = 3, biological replicates). Total cell number was determined through a DNA intercalating dye.

**Figure 2 biomedicines-10-00266-f002:**
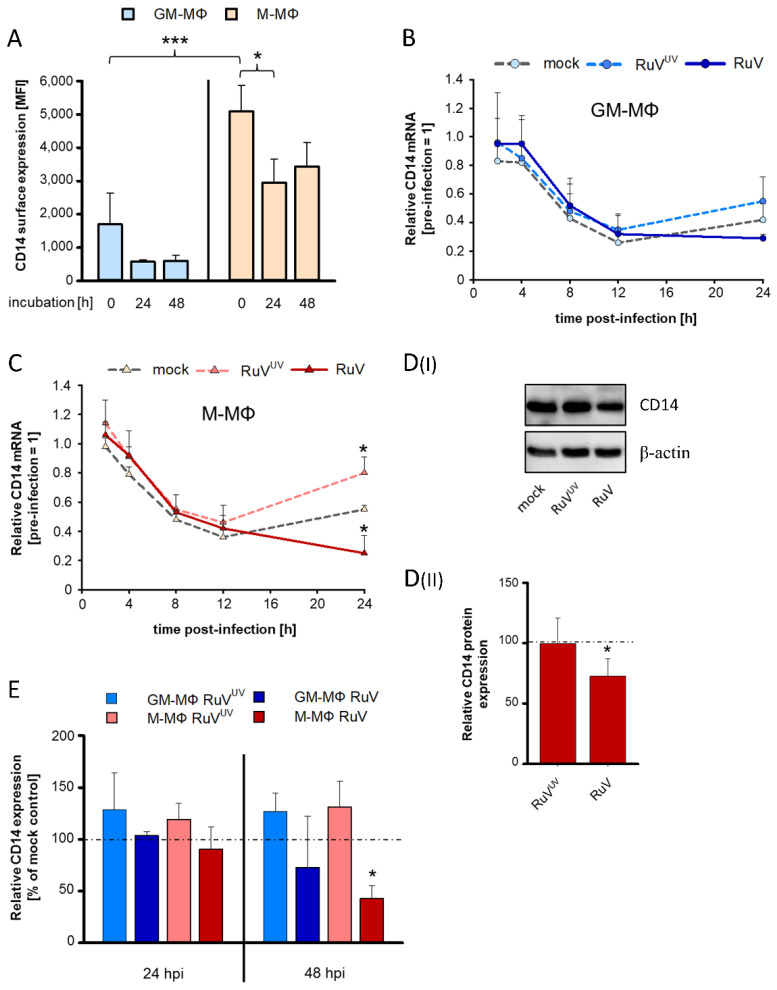
Downregulation of CD14 expression in GM- and M-MΦs after RuV infection. (**A**) Surface expression of CD14 on uninfected MΦ was determined by flow cytometric analysis for indicated time points after plating of MΦs. Data represent mean fluorescence intensity (MFI) ± SD (*n* = 3). Statistical analysis was performed using ANOVA test. (**B**,**C**) At 2, 4, 8, 12, and 24 hpi CD14 mRNA level was quantified by qPCR relative to the before infection control (0 hpi = 1). Statistical analysis was performed using Student’s *t*-test and significance was calculated to the respective mock control. (**DI**,**DII**) RIPA lysates were generated from RuV- and RuV^UV^-infected MΦs at 24 hpi and analysed by Western blot with anti-E1 and β-actin antibodies. (**DI**) One representative blot out of three is shown. (**DII**) Western blot bands were quantified and E1 protein was normalized to loading control β-actin. Data represent means ± SD (*n* = 3). Statistical analysis was performed using ANOVA test and significance was calculated to the mock control. (**E**) At 24 and 48 hpi CD14 surface expression on RuV- and RuV^UV^-infected MΦ was determined by flow cytometric analysis. Percent values of MFI ± SD are shown (*n* = 3) and expressed as percent change compared to mock control (=100%). Student’s *t*-test was performed for statistical analysis and significance was calculated to the mock control. * *p* ≤ 0.05, *** *p* ≤ 0.001.

**Figure 3 biomedicines-10-00266-f003:**
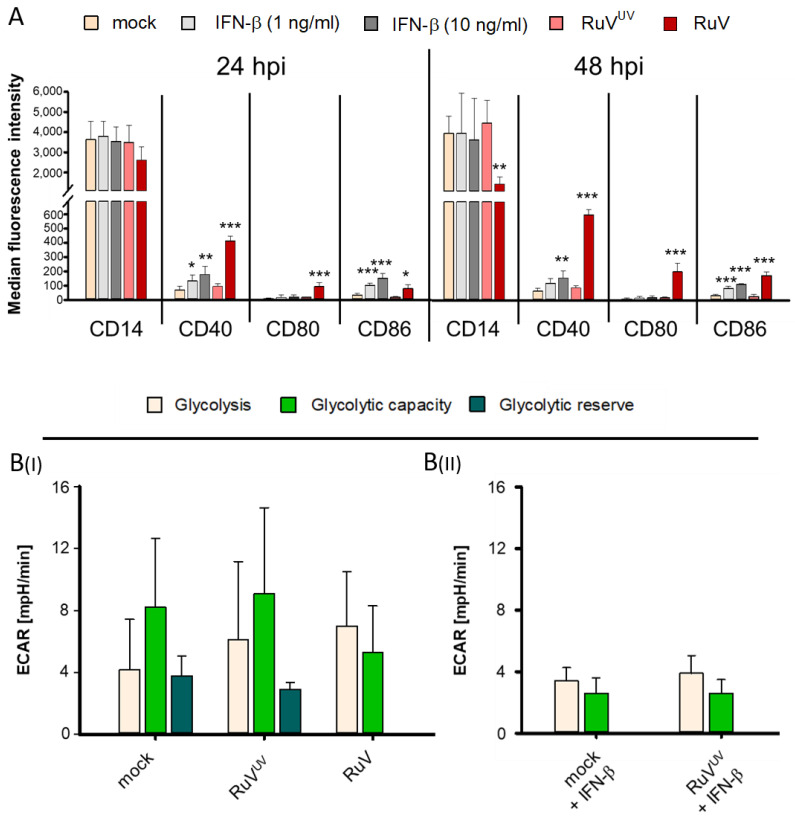
During RuV infection and in the presence of exogenous IFN-β glycolytic reserve was lost in M-MΦ. (**A**) Analysis of surface expression of CD14, CD40, CD80, and CD86 in mock-, RuV^UV^-, and RuV-infected and exogenous IFN-β treated GM- and M-MΦ by flow cytometry after indicated incubation intervals. Relative surface marker expression after application of exogenous IFN-β (10 ng/mL) for 24 h was assessed by flow cytometry with the respective antibodies. Values (*n* = 3) of the isotype-corrected MFI ± SD are shown. Statistical analysis was performed using ANOVA test and significance was calculated to the respective mock control. (**B**) M-MΦ (5 × 10^5^/mL) were (**BI**) mock-infected or infected with RuV^UV^ or RuV or (**BII**) mock- and RuV^UV^-infected in the presence of IFN-β (10 ng/mL). Glycolysis was measured through extracellular flux analysis as real-time ECAR by the glycolysis stress test and successive injection of glucose to measure glycolysis and of oligomycin to measure glycolytic capacity and reserve. Data are shown in bars as the mean plus standard deviation (±SD) and were derived from three independent experiments with two wells per sample and experiment. * *p* ≤ 0.05, ** *p* ≤ 0.01, *** *p* ≤ 0.001.

**Figure 4 biomedicines-10-00266-f004:**
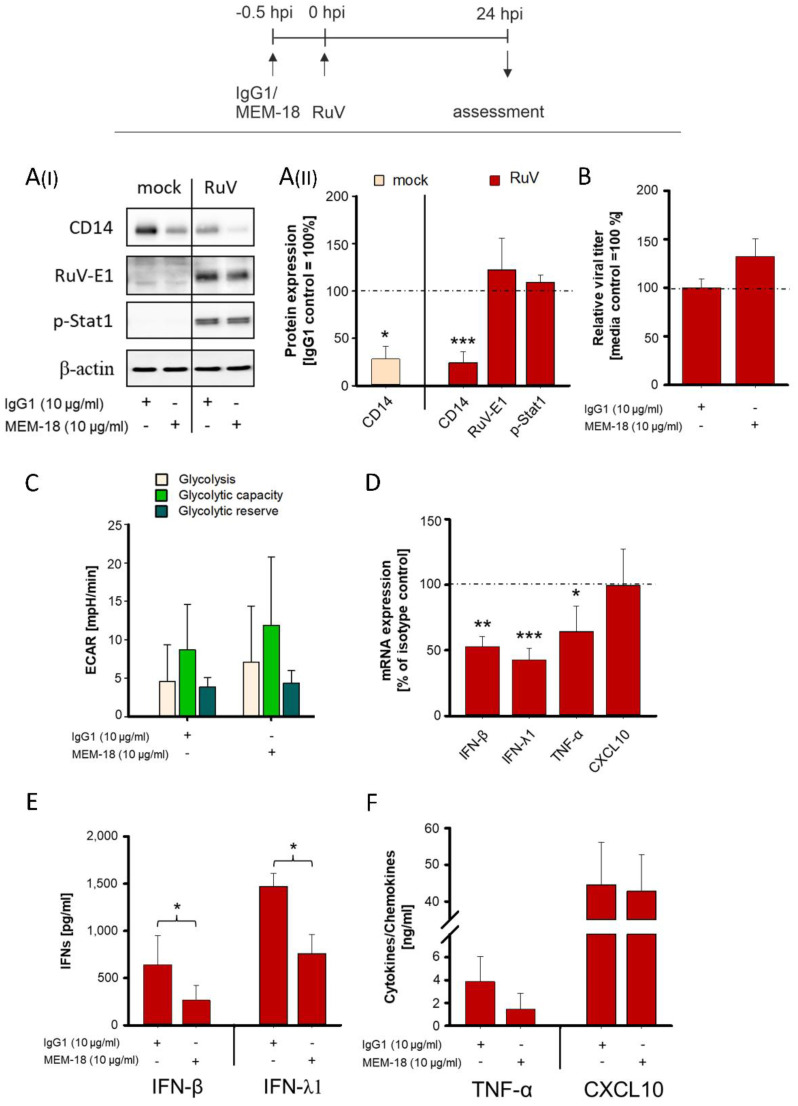
The application of blocking antibodies prior to RuV infection of M-MΦ reveals CD14-dependent and -independent mechanisms of the associated inflammatory response. Anti-CD14 blocking or corresponding isotype control antibodies were applied to M-MΦ (5 × 10^5^/mL) 30 min before control infection or infection with RuV. (**A**) Western blot analysis of protein lysates (30 µg/lane) with anti-CD14, anti-E1 and anti-β-actin antibodies at 24 hpi. (**AI**) One representative Western blot from independent experiments of four donors is shown. (**AII**) Western blot results were subjected to densitometric analysis for quantification of expression changes (*n* = 4). Statistical analysis was performed using ANOVA test and significance was calculated to the respective isotype control (**B**) Extracellular virus particles were determined at 24 hpi by plaque assay. Percent values of the means ± SD are shown (*n* = 3) and expressed as percent change compared to isotype control. (**C**) M-MΦs were subjected to extracellular flux analysis with the glycolysis stress test 24 h after addition of the blocking and control antibodies. Glycolysis and glycolytic capacity/reserve were measured through successive injection of glucose and oligomycin, respectively. Data are shown in bars as the mean plus standard deviation (±SD) and were derived from three independent experiments with two wells per sample and experiment. (**D**) The mRNA expression of the indicated cytokines/chemokines in RuV-infected MΦs was determined at 24 hpi by quantitative real-time PCR. Data normalization was carried out with GNB2L1. Relative mRNA quantification based on 2^−∆∆Ct^ was employed. Expression of indicated target mRNA was determined relative to isotype control (isotype control = 100%) ± SD (*n* = 3). Statistical analysis was performed using ANOVA test and significance was calculated to the isotype control. (**E**,**F**) The protein concentrations of the indicated cytokines/chemokines in supernatants of RuV-infected MΦs were determined at 24 hpi by LEGENDPLEX human interferon panel kit (IFN-β, IFN-λ1 and TNF-α) or ELISA (CXCL10). Data are shown in bars as the mean plus standard deviation (±SD) (*n* = 3). Statistical analysis was performed using ANOVA test and significance was calculated to the respective isotype control. * *p* ≤ 0.05, ** *p* ≤ 0.01, *** *p* ≤ 0.001.

**Figure 5 biomedicines-10-00266-f005:**
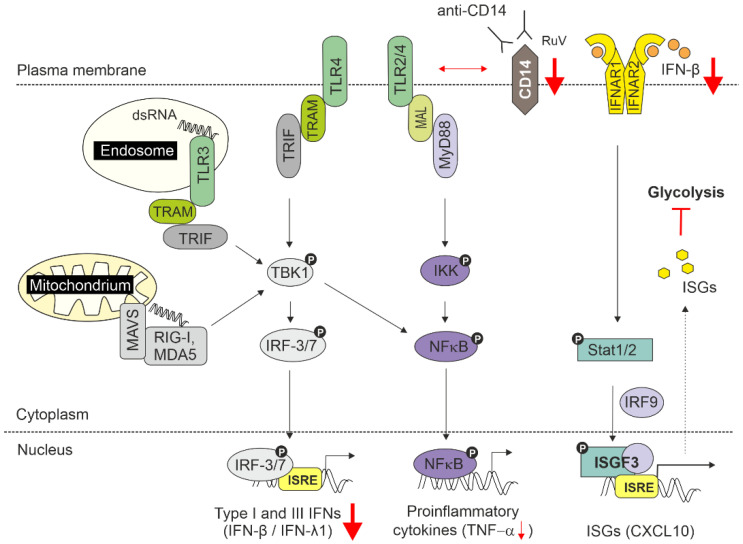
Summarizing figure. CD14-dependent signalling contributes to the induction of IFNs during RuV infection and potentially involves a so far unidentified TLR. As recently outlined in a review, among the TLRs involved in recognition of viral PAMPs are TLR2 and TLR4 (envelope glycoproteins) and TLR3 (dsRNA) [38]. For simplification, internalization and localization of TLR2 and TLR4 to endosomes are not depicted. All TLRs use the adaptor MyD88 except for TLR3, which signals only through TRIF. TLR3 and TLR4 interact with the adaptor TRIF. As the PRR for RuV is still unknown, RIG-I and MDA5 are included as additional PRRs for dsRNA within the cytoplasm. The associated transcriptional pathways, NFkB and IRF3/7, are illustrated. Through its association with TLRs, CD14 modulates the activation of downstream signalling pathways. The noted reduction in IFNs after application of blocking antibodies prior to RuV infection points towards the involvement of TRIF-dependent pathways during RuV infection. Downmodulated targets are highlighted with red arrows. The inhibitor symbol indicates the here identified inhibition of glycolysis by IFN-β.

## Data Availability

The raw data supporting the conclusions of this article will be made available by the authors, without undue reservation.

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
