# Peer review of "CD14 Is Involved in the Interferon Response of Human Macrophages to Rubella Virus Infection"

_biomedicines, 2022, doi:10.3390/biomedicines10020266_

Round 1

Reviewer 1 Report

This is the manuscript on the observed downmodulation of CD14 during rubella virus infection of anti-inflammatory M2-like appears to contribute to virus-host-adaption through a reduction of the IFN response. Authors use the method that blockage of CD14 increased the amounts of extracellular virus particles and altered specific components of the cytokine/chemokine profile after infection with rubella virus. The manuscript is well written but requires some revisions as follows.

 Revision points

 1: Authors described that in response to RuV infection, GM- and M-MΦ generate type I and III IFN (page 9 line286). This may be a known fact among experts. But I think showing the upregulation of INF-β(protein or mRNA level) post-infection to M-MΦ in the figure is important to understand this mechanism as well as citing the paper. It is easier for readers to understand the mechanism of the complicated mechanism of RuV and CD14 and INF-β’s relationship.

2: In Figure 3A, the authors described that “RuVUV infection altered only CD14 expression, which was increased after 48 hours of incubation “ (page 9 line 300). Is it a significant difference? Please clarify.

3: On page 9 line 308-309, the authors described that “after RuV infection glycolytic capacity was slightly reduced in comparison to the mock control…”.  Can you define “the slightly reduced”? And authors mentioned that “glycolytic reserve was not measurable”. Is it a significant difference between mock group and RuV group?

 4: In Figure4A(II), the number of experiments is not written. Is it three times? And I do not understand the meaning of “protein expression/phosphorylation” of the vertical axis.

Author Response

Reviewer 1 – revision points

The authors want to thank the reviewer for the time spent on our manuscript and the helpful comments.

Point-by-point response to the reviewer’s suggestions

1: Authors described that in response to RuV infection, GM- and M-MΦ generate type I and III IFN (page 9 line286). This may be a known fact among experts. But I think showing the upregulation of INF-β (protein or mRNA level) post-infection to M-MΦ in the figure is important to understand this mechanism as well as citing the paper. It is easier for readers to understand the mechanism of the complicated mechanism of RuV and CD14 and INF-β’s relationship.

Response: We have revised the original sentence (page 9 line 286) and expanded the statement as follows (page 9 line 292):

“The infection with RuVUV induced no IFN-ß production (Schilling et al. 2021, DOI: 10.3389/fimmu.2021.772595). Our own published data shows that at 24 hpi IFN-ß as an antiviral response to RuV infection was generated on GM- and M-MΦ at a concentration of 219 ± 170 pg/ml and 666 ± 199 pg/ml, respectively (Schilling et al. 2021, DOI: 10.3389/fimmu.2021.772595). This is also illustrated in Supplement Figure 2, which indicates the generation of IFN-ß at a concentration of 466 ± 119 pg/ml on M-MΦ at 24 hpi. After medium change performed at 24 hpi IFN-ß dropped to a concentration of 59 ± 38 pg/ml after an additional incubation of 24 hours. In order to investigate which effects of the RuV infection on M-MΦ are associated with IFN-ß production, exogenous IFN-ß was applied to M-MΦ and compared to RuV infection.”

 2: In Figure 3A, the authors described that “RuVUV infection altered only CD14 expression, which was increased after 48 hours of incubation“ (page 9 line 300). Is it a significant difference? Please clarify.

Response: We understand the reviewer's objection, since the slight shift in CD14 surface expression under RuVUV infection was not a significant change compared to the mock control. We changed this sentence as follows (page 9 line 314):

“Under these conditions, the influence of RuVUV infection on the surface expression of the examined markers was not significant in comparison to the mock control. (Figure 3A).”

3: On page 9 line 308-309, the authors described that “after RuV infection glycolytic capacity was slightly reduced in comparison to the mock control…”.  Can you define “the slightly reduced”? And authors mentioned that “glycolytic reserve was not measurable”. Is it a significant difference between mock group and RuV group?

Response: As the reviewer noted, the shift in the glycolytic reserve in RuV infection was not a significant change compared to the mock control. The main observation from the experiment is the loss of the glycolytic reserve after RuV infection. In the mock control the glycolytic reserve is present, but disappears completely and was detected at the value zero after RuV infection.

We rephrased the sentence as follows (page 9 line 322):

“Figure 3B(I) underlines that although after RuV infection glycolysis and glycolytic capacity did not shift significantly compared to the mock control; the glycolytic reserve changed significantly and was no longer measurable.”

4: In Figure4A(II), the number of experiments is not written. Is it three times? And I do not understand the meaning of “protein expression/phosphorylation” of the vertical axis.

Response: Thank you for pointing this missed aspect out. We have added to the figure legend (Figure 4A(II)) that it is a representative blot from independent experiments of four donors (n = 4).

Following the reviewer's comment, the y-axis title was changed to “protein expression [IgG1 control = 100%]”.

Reviewer 2 Report

Schilling et al. reported that CD14 is involved in the IFN response of human macrophages to rubella virus infection. These observation seem important findings in this area.

  1. In Introduction section, please mention about rubella virus and the association between rubella virus and macrophages.
  2. The RuV genome is a 9.7-kb linear single-stranded RNA of positive polarity, which encodes for three structural proteins (envelope glycoproteins E1 and E2, and capsid protein C) and two nonstructural replicase proteins p150 and p90 [Lambert N, Strebel P, Orenstein W, et al. Rubella. Lancet 2015; 385:2297–2307.]. How the pattern recognition receptors? As its pathogen-associated molecular patterns are RNAs, are PRRs MDA5, RIG-I or TLR3? Authors should add these molecules and TLR4 in Figure 4. Authors should discuss more about them.
  3. Did you use the siRNA against CD14, TLR4 or both?

Author Response

#Reviewer 2 – revision points

The authors want to thank the reviewer for the time spent on our manuscript and the helpful comments.

Point-by-point response to the reviewer’s suggestions

  1. In Introduction section, please mention about rubella virus and the association between rubella virus and macrophages.

Response: We have inserted the following text passage (page 2 line 54):

“There is an association between RuV and MΦ as these cells play an important role in the replication and spread of RuV and as such in rubella pathogenesis. In line with this, van der Logt et al. showed in an early work that monocyte-derived MΦ support RuV replication (van der Logt et al. 1980, DOI: 10.1128/iai.27.2.309-314.1980). Furthermore, the results of Lazar et al. revealed that in tissue samples from patients with fatal congenital rubella syndrome (CRS) mainly alveolar MΦ were positive for RuV antigen (Lazar et al. 2016, DOI: 10.1016/j.ebiom.2015.11.050). In our previous work we demonstrated that both human GM-MΦ and human M-MΦ were infected with RuV and that replication of RuV occured at a similar rate in both cell types (Schilling et al. 2021, DOI: 10.3389/fimmu.2021.772595). 

  1. The RuV genome is a 9.7-kb linear single-stranded RNA of positive polarity, which encodes for three structural proteins (envelope glycoproteins E1 and E2, and capsid protein C) and two nonstructural replicase proteins p150 and p90 [Lambert N, Strebel P, Orenstein W, et al. Rubella. Lancet 2015; 385:2297–2307.]. How the pattern recognition receptors? As its pathogen-associated molecular patterns are RNAs, are PRRs MDA5, RIG-I or TLR3? Authors should add these molecules and TLR4 in Figure 4. Authors should discuss more about them.

Response: We have expanded and revised Figure 4 in accordance with the reviewer's suggestions and added the following text to the Figure legend (page 13 line 420):

“Among the TLRs involved in recognition of viral PAMPs are TLR2 and TLR4 (envelope glycoproteins) and TLR3 (dsRNA) (Sartorius et al. 2021, DOI: 10.1038/s41541-021-00391-8). For simplification, internalization and localization of TLR2 and TLR4 to endosomes are not depicted. All TLRs use the adaptor MyD88 except for TLR3, which signals only through TRIF. TLR3 and TLR4 interact with the adaptor TRIF. As the PRR for RuV is still unknown, RIG-I and MDA5 are included as additional PRRs for dsRNA within the cytoplasm.”

The additions to Figure 4 were also taken up in the discussion section and the following text passage (page 14 line 454) was added:

“The RuV PAMPs and the receptors involved in their recognition are still not known. Nevertheless, human nucleotide polymorphisms (SNPs) that are associated with variations in the immune response to rubella vaccination provide some indications. The cytokine response to RuV vaccine strain including the generation of IFN-γ, TNF-α, and GM-CSF were affected by SNPs in the IFN-α/β receptor (IFNAR2) and the TLR3 and TLR4 genes (Ovsyannikova et al. 2014, DOI: 10.1007/s00251-014-0796-z), (Ovsyannikova et al. 2015, DOI: 10.1007/s00251-015-0864-z), (Ovsyannikova et al. 2010, DOI: 10.1007/s00439-009-0763-1). Thus, the identified SNPs could be involved in a differential immune response to rubella vaccination in humans. Namely, two SNPs within the TLR3 gene were accompanied by a reduced secretion of GM-CSF (Ovsyannikova et al. 2010, DOI: 10.1007/s00439-009-0763-1). Besides the contributory role of the PRRs TLR3 and TLR4, these studies also highlight the importance of innate immune responses for the overall antiviral countermeasures against RuV. The generation of dsRNA during RuV infection appears to be the main activating signal for an IFN response as RuVUV does hardly induce IFNs (Schilling et al. 2021, DOI: 10.3389/fimmu.2021.772595).”

  1. Did you use the siRNA against CD14, TLR4 or both?

Response: For the work presented here, we did not use siRNAs for CD14 or TLR4. MΦ are not easy to target experimentally by RNA or DNA transfection. Thus, THP-1 cells are generally used as a model system. Although they resemble human MΦ, they also differ at important aspects. This includes a low expression level of CD14 on THP-1 cells. Another aspect that argues against the application of siRNAs: CD14 and TLR4 are two differentiation markers for human MΦ and are thus essential for the characteristics of these cells.

We have established a functional system for the transfection of siRNAs in MΦ in our laboratory (unpublished data), we tried to transfer this system to CD14, but we could only achieve a slight downregulation at mRNA level, while protein expression remained nearly unchanged. This can also be due to the fact that CD14 protein on the surface of MΦ does not undergo a large turnover/expression without additional stimulation.

The TLR4 knockdown in human MΦ would present us with similar challenges with the additional fact that TLR4 is probably only slightly expressed on the cell surface, since the marker is very difficult to detect in flow cytometry (our own unpublished observations).